# Increased Motility in *Campylobacter jejuni* and Changes in Its Virulence, Fitness, and Morphology Following Protein Expression on Ribosomes with Altered RsmA Methylation

**DOI:** 10.3390/ijms25189797

**Published:** 2024-09-10

**Authors:** Agnieszka Sałamaszyńska-Guz, Małgorzata Murawska, Paweł Bącal, Agnieszka Ostrowska, Ewelina Kwiecień, Ilona Stefańska, Stephen Douthwaite

**Affiliations:** 1Division of Microbiology, Department of Pre-Clinical Sciences, Institute of Veterinary Medicine, Warsaw University of Live Sciences—SGGW, Ciszewskiego 8, 02-786 Warsaw, Poland; malgorzata_murawska@sggw.edu.pl (M.M.); ewelina_kwiecien1@sggw.edu.pl (E.K.); ilona_stefanska@sggw.edu.pl (I.S.); 2Institute of Paleobiology, Polish Academy of Sciences, Twarda 51/55, 00-818 Warsaw, Poland; bacal@twarda.pan.pl; 3Department of Nanobiotechnology, Institute of Biology, Warsaw University of Live Sciences—SGGW, Ciszewskiego 8, 02-786 Warsaw, Poland; agnieszka_ostrowska@sggw.edu.pl; 4Department of Biochemistry and Molecular Biology, University of Southern Denmark, Campusvej 55, DK-5230 Odense, Denmark

**Keywords:** *Campylobacter*, flagellar synthesis, biofilm formation, epithelial cell invasion, rRNA methylation, proteomics

## Abstract

Infection with *Campylobacter jejuni* is the major cause of human gastroenteritis in the United States and Europe, leading to debilitating autoimmune sequelae in many cases. While considerable progress has been made in detailing the infectious cycle of *C. jejuni*, a full understanding of the molecular mechanisms responsible for virulence remains to be elucidated. Here, we apply a novel approach by modulating protein expression on the pathogen’s ribosomes by inactivating a highly conserved rRNA methyltransferase. Loss of the RsmA methyltransferase results in a more motile strain with greater adhesive and cell-invasive properties. These phenotypical effects correlate with enhanced expression of specific proteins related to flagellar formation and function, together with enzymes involved in cell wall/membrane and amino acid synthesis. Despite the enhancement of certain virulent traits, the null strain grows poorly on minimal media and is rapidly out-competed by the wild-type strain. Complementation with an active copy of the *rsmA* gene rescues most of the traits changed in the mutant. However, the complemented strain overexpresses *rsmA* and displays new flaws, including loss of the spiral cell shape, which is distinctive for *C. jejuni*. Proteins linked with altered virulence and morphology are identified here by mass spectrometry proteomic analyses of the strains.

## 1. Introduction

*Campylobacter* infections are the leading cause of human bacterial gastroenteritis. Since 2005, campylobacteriosis has been the most commonly reported zoonosis in the European Union, representing more than 60% of all cases in 2022 (EFSA, ECDC 2023). An acute infection can have serious long-term sequelae, including peripheral neuropathies, Guillain–Barré syndrome (GBS), Miller Fisher syndrome (MFS), and irritable bowel syndrome [1]. The virulence mechanisms of *Campylobacter jejuni,* the most prevalent member of this genus of pathogens, are still not fully understood.

In one approach to gain insight into these mechanisms, we previously showed that a specific modification of the ribosomal RNA (rRNA) of *C. jejuni* modulates its virulence characteristics [2]. Modification of *C. jejuni* 23S rRNA on the 2′-*O*-position of nucleotide C1920 is catalyzed by the methyltransferase TlyA, and inactivation of this enzyme results in a wide range of effects, including defective ribosomal subunit association and reduced motility, biofilm formation, adherence/invasion into human epithelial cells, and survival within macrophages [2,3]. Loss of the TlyA methylation and the resultant phenotypic effects correlate with specific changes in the protein composition of *C. jejuni* [4]. The rRNAs of all bacteria contain numerous modifications, for the most part, methylations, that are added to specific nucleotides by individual enzymes [5,6,7]. Many of these nucleotide methylations are highly conserved phylogenetically. It is thus likely that TlyA would not be the only rRNA methyltransferase to affect the content of the bacterial proteome and that other rRNA methylations could modulate protein synthesis in subtly different ways. The effects of these changes in the proteome would, in turn, give further insight into how the levels of expression of specific proteins contribute to *C. jejuni* virulence.

In the present study, we focus on the methyltransferase RsmA (formerly KsgA), which is universally conserved amongst bacteria, Archaea, and Eukarya [8] and is thus more prevalent than TlyA. RsmA catalyzes *N6,N6*-dimethylation at the two invariant adenosines A1518 and A1519 (*Escherichia coli* numbering) within the loop of helix 45 at the 3΄-end of the small ribosomal subunit RNA [9]. *E. coli rsmA*-null mutants show resistance to the antibiotic kasugamycin [10] and are prone to decoding errors, exhibit a cold-sensitive phenotype, and display an overall loss of fitness [11,12]. Similar to both Gram-positive and Gram-negative bacterial pathogens, inactivation of RsmA is accompanied by a decrease in virulence potential. For instance, survival of a RsmA-deficient mutant of *Yersinia pseudotuberculosis* is significantly impaired in both a murine model of infection and in cultured HeLa cells [13,14], and deletion of RsmA in *Salmonella enterica* serovar Enteritidis reduces invasiveness in cultured human intestinal epithelial cells and chicken liver cells and reduces survival in egg albumen [15]. RsmA has also been shown to contribute to virulence in the Gram-positive pathogen *Staphylococcus aureus* [16]. Despite the obvious importance of RsmA in protein synthesis, no organism, to our knowledge, has previously been analyzed to determine how this enzyme affects the contents of a cell’s proteome.

Here, we have inactivated the *rsmA* gene of *C. jejuni*. As would be predicted from the studies on other bacterial pathogens, loss of the RsmA enzyme and resultant lack of methylation at rRNA nucleotides A1518 and A1519 confers increased tolerance to kasugamycin. Inactivation of *rsmA* in *C. jejuni* results in the attenuation of some virulence traits while enhancing several others, resulting in a phenotype that is distinctly different from that of the *tlyA* mutant. Mass spectrometric analyses of the proteomes of the *C. jejuni* mutant strain, before and after complementation with an active *rsmA* gene, give fresh insight into how biofilm formation, motility, epithelial cell invasion, and cell morphology are linked to the expression levels of specific *C. jejuni* proteins.

## 2. Results

### 2.1. Changes in the C. jejuni Growth Phenotype after Loss of RsmA Methylation

Disruption of the *C. jejuni rsmA* gene by insertion of the *cat* cassette (Figure 1A) was confirmed by PCR sequencing of the chromosome region and by the acquisition of chloramphenicol resistance; the resultant lack of methylation at 16S rRNA nucleotides A1518 and A1519 was verified by primer extension (Figure 1B). Methylation of the rRNA was reestablished after complementing the null strain with an active copy of *rsmA*.

No difference was detected in the growth rates of the *C. jejuni* wild-type, null mutant, and complemented strains in rich medium (MH broth) at 37 °C (Figure 2A,C). However, as predicted from *rsmA* studies in other bacteria, inactivation of this gene conferred a growth advantage to the null mutant in the presence of the antibiotic kasugamycin. The *rsmA* null strain grew comfortably at 1600 µg kasugamycin mL^−1^ MH broth, whereas the wild-type and complemented strains did not grow at 800 µg kasugamycin mL^−1^ (Figure 1C). Consistent with this, growth of the null strain in a medium with sub-inhibitory concentration (400 µg mL^−1^) of kasugamycin was faster than the wild-type and complemented strains (Figure 2A), and in direct competition assays with this amount of drug, the wild-type cells were barely detectable after the second growth cycle (Figure 2C). Interestingly, the complemented strain was consistently more sensitive to kasugamycin than the original wild-type strain (Figure 1C).

Although no changes in the growth of the *rsmA* null strain were detected in a rich medium without a drug, defects became evident when grown in minimal media. In the MCLMAN minimum medium and particularly in the MCLAN medium (lacking methionine), the growth of the *rsmA* mutant was markedly weaker than the wild-type strain (Figure 2D,E). In competition assays, the proportion of mutant cells gradually declined and disappeared from the total population after three growth cycles in the MCLMAN medium and after two cycles in MCLAN (Figure 2F). The growth rate of the complemented strain was restored in the MCLAN medium but not in the MCLMAN medium, where growth remained at the level of the null mutant.

### 2.2. Changes in C. jejuni Biofilm Formation after Loss of RsmA Methylation

The ability of the *rsmA*-null strain to form biofilm on a polystyrene surface determined by the crystal violet assay was increased relative to the wild-type strain (Figure 3A,B). Complementation of the null strain with a functional copy of the *rsmA* gene reduced poly-styrene adherence to the level observed in the wild-type strain (Figure 3A,B). However, the reverse of these effects was observed in the MCLMAN and MCLAN minimal media, where the *rsmA* strain showed reduced biofilm-forming ability compared to the wild-type and the complemented strains (Figure 3C,D).

### 2.3. Changes in C. jejuni Virulence after Loss of RsmA Methylation

Surprisingly, the *rsmA* null mutant exhibited increased motility with a movement zone on agar plates that was almost twice as large as that of the wild-type. This effect was eliminated and restored to wild-type levels after complementation with an active copy of *rsmA* (Figure 4A). Scrutinizing the cells by electron microscopy, SEM showed that while almost all mutant cells possessed flagella, significantly fewer flagellated cells were observed for the wild-type and complemented strains (Figure 4B). The length of the flagella was similar in all strains. Observation of cells under phase contrast microscopy revealed that wild-type *C. jejuni* cells tended to tumble and travel for shorter distances in contrast to the *rsmA* mutant, where many cells exhibited a darting motility and traveled further with greater velocity (videos in the Appendix A).

Coinciding with the increase in motility, the internalization ability of the *rsmA* mutant into epithelial cells was enhanced about twenty-fold in comparison with the wild-type strain. Complementation of the mutant with an active *rsmA* gene annulled this effect (Figure 5A). As a consequence of its greater internalization ability, a larger proportion of *rsmA* mutant bacteria were initially detected in RAW264.7 macrophages compared to the wild-type and complemented strains (Figure 5B). However, despite the entry of the larger number of the *rsmA* cells, combined with their longevity in the macrophages being marginally longer than that of the wild-type and complemented strains, none of the internalized strains were found to survive more than 48 h.

### 2.4. Upregulated Expression of Proteins Associated with Motility in the C. jejuni rsmA Mutant

The genome of *C. jejuni* 81-176, together with its two resident plasmids, encodes 1813 open reading frames (ORFs). Using a mass spectrometry-based quantitative proteomic approach, we detected 1307 of these protein products (1246 encoded by the chromosome, 29 by plasmid pVir, and 32 by plasmid pTet), representing 69% of the potential proteome.

Deletion of the *rsmA* gene in *C. jejuni* 81-176 altered the abundance of 147 proteins compared to the wild-type strain, 81 of which were upregulated and 66 downregulated (Appendix A). As expected, a functional RsmA methyltransferase was absent in the mutant strain (although, because the *rsmA* gene was inactivated but not removed, peptides corresponding to RsmA were detected in the analyses (Appendix A)). Complementing this strain with an active copy of *rsmA* reduced the number of significant changes to 30 proteins (Table 1 and Appendix A). These changes included several major flagellar components, including FlaA, FlaB, and FliD; the main hook protein FlgE, the basal body proteins FlgG; the flagellar type III secretion system (T3SS) FlhA and FlaG; PseA (two-fold-higher levels than in the wild-type) and Cjj0996 (three-fold higher); and also FlgP (four-fold-higher levels) (Figure 6A and Table 1). Similar increases were seen in the abundances of the phosphate transporter PstS and the histidine synthesis enzymes HisF2 and HisH1. Other proteins classed as significantly changed by lack of RsmA methylation include the ribosomal protein S12, PepP, Cjj1105, Cjj1656, and three proteins encoded by pTet plasmid. These amounts of these latter proteins doubled upon *rsmA* inactivation, and although their levels were significantly rescued by complementation, they remain slightly higher than in the wild-type strain (Table 1 and Appendix A).

Downregulated proteins included those involved in amino acid transport (GlnH), pseudouridine synthase RluB, and seven plasmid-encoded proteins. The abundances of all these proteins returned to almost wild-type levels after complementation (Appendix A). However, several proteins were not rescued (partially or completely) to wild-type levels by complementation. This was possibly a consequence of secondary site mutations or polar effects associated with mutagenesis rather than being directly caused by the loss of the RsmA protein. This is clear, for instance, in the case of Cjj0003 (RluB) and Cjj0004, which are encoded immediately downstream of *rsmA* and were presumably affected by the upstream insertion of the *cat* cassette.

## 3. Discussion

Nucleotide modifications fine-tune the functional roles of rRNAs in ribosome biogenesis, in the translational process from initiation to termination, and in resistance to antibiotics [17,18]. Altering rRNA modifications can affect the translation process in a way that skews the relative proportions of certain proteins [4], which in turn results in physiological changes, such as the attenuation of specific virulence properties in bacterial pathogens [2,13,14,15,16,19]. Changes in protein expression might be expected to be most pronounced when a phylogenetically conserved modification enzyme is inactivated (or perhaps over-expressed), and here we show that carrying out a full proteomic analysis of a strain lacking such a modification can lead to the identification of protein factors that are important for pathogenicity.

In this study, we have inactivated the *C. jejuni* enzyme RsmA, which is a universally conserved dimethyltransferase responsible for the post-transcriptional modification of nucleotides A1518 and A1519 located near the 3′ end of 16S rRNA [20,21]. Dimethylation of the small ribosomal subunit at this location is highly conserved, with homologs of RsmA present in all free-living organisms and most organelles in the three domains of life [5]; only a few intracellular bacterial parasites and commensal Nanoarchaeota have been found to lack an RsmA homolog and the associated rRNA methylations [22]. Inactivation of RsmA in bacteria confers resistance to the antibiotic kasugamycin [10], and mutation of the rRNA indicates that nucleotide A1519 is the more important determinant of resistance [23]. In this context, we note that the kasugamycin resistance conferred by the inactivation of *rsmA* in *C. jejuni* was not only annulled by complementation with an active copy of *rsmA*, but the complemented strain became even more drug-sensitive than the wild-type strain (Figure 2A).

The primer extension assay clearly shows that nucleotide A1519 is fully dimethylated in the rRNAs from both the wild-type and complemented strains (Figure 1B), but (due to the reverse transcriptase stop at A1519) this method gives no indication of the methylation status at A1518. From the proteomics data (Appendix A), expression of the RsmA methyltransferase in the complemented strain can be seen to be about three times the level in the wild-type strain, despite transcription of *rsmA* from its own promoter, and this effect must, therefore, be due to the displaced location of the gene in the chromosome (Figure 1A). Given the discrepancies in RsmA expression, it is plausible that A1518 is undermethylated in the wild-type strain and more completely dimethylated in the complemented strain, and this would account for the increased kasugamycin sensitivity (Figure 1C), and possibly also for some of the protein expression effects where complementation overcompensated their rescue (discussed below).

Aside from this well-documented role of RsmA in kasugamycin resistance, the main purpose of this study was to investigate how the lack of this enzyme alters the virulence potential of *C. jejuni*. One of the most noticeable physiological changes in the *rsmA* mutant is its increased motility in a rich medium, which is evident both on agar (Figure 4A) and in liquid culture (videos in the Appendix A). This effect coincides with the expression of higher levels of the main functional and structural elements of the two polar flagella (Figure 6A), which, in combination with the helical cell body of *C. jejuni*, facilitate mobility through highly viscous environments [24]. FlaA is a major flagellin protein that, together with FlaB, is essential for the formation of the full-length flagellar filament [25], while FlaG controls the length of the flagella [26]. Each of these proteins is expressed at two to three times higher levels in the *rsmA* mutant (Figure 6A). FlgP (four-fold overexpressed in the *rsmA* strain) is mainly located in the outer membrane, and its absence results in flagellar paralysis [27,28].

From viewing electron micrographs of several hundred individual cells from each of the strains, we can conclude that whereas the *rsmA* mutant cells in nearly all cases displayed full-length flagella, no flagella could be detected in approximately half of the wild-type cells (Figure 4B). In the complemented strain, flagella were slightly more prevalent than in the wild-type population of cells, but their cell morphology was distinctly different. Typically, wild-type *C. jejuni* cells have a helical appearance, and this shape is retained in the *rsmA* strain. However, upon complementation, the cells lost curvature and became rod-shaped (Appendix A). More complete methylation of the 16S rRNA by more abundant copies of RsmA is presumably affecting the expression of factors influencing cell morphology, and likely candidates here are Cjj1105, the transpeptidase PbpC, and the peptidoglycan peptidase Pgp2, where the latter two proteins are markedly less plentiful in the complemented strain (Appendix A). Cjj1105 [29] and Pgp2 [30] have previously been associated with *C. jejuni* cell curvature, and PbpC has previously been associated with cell elongation [31].

The multitude of effects seen here in protein expression (147 significant changes, Appendix A) are certainly not all a direct result of RsmA inactivation. This number was reduced to 30 significant changes by comparison of the *rsmA* strain with the complemented strain, but even so, some of these changes will be an indirect consequence of *rsmA* inactivation. For instance, the expression of flagellar genes is hierarchically controlled to facilitate the ordered secretion of flagellar proteins, such that their biosynthesis proceeds in a sequential manner. The increased abundance of FlhA in the *rsmA* mutant (Figure 6A) would enhance the expression of subsequent proteins in the flagellar biosynthesis hierarchy (Figure 6B), presumably through σ^54^ and σ^28^ regulation [32]. FlhA is an essential part of the T3SS secretion system and, together with the FlgSR two-component system, plays a key role in the flagellar export apparatus of *C. jejuni* [33]. In other bacterial pathogens, including *Bacillus thuriengensis* [34], *Helicobacter pylori* [35], and *Pseudomonas aeruginosa* [36], FlhA has been shown to play a major role in the regulation of flagellar expression and/or virulence.

In addition to motility, the flagella of *C. jejuni* are important for biofilm formation and for the invasion of epithelial cells [37,38]. Both of these traits are key aspects of *C. jejuni* virulence, and taking biofilm formation first, the *rsmA* mutant clearly has an enhanced ability to adhere to surfaces (Figure 3A,B). Consistent with this observation, the *C. jejuni* flagellar filaments FlaA, FlaB, and FlaG, the hook protein FlgE, and the T3SS protein FlhA have previously been shown to enhance biofilm formation [39,40,41,42].

In the case of epithelial cell invasion by *C. jejuni,* the disproportionally great increase in the ability of the *rsmA* mutant to be internalized was unexpected (Figure 5A) and thus at odds with observations made for *S.* Enteritidis [15]. The twenty-fold greater invasion of epithelial cells by the *C. jejuni rsmA* strain might not be readily explained merely by the increases in proteins involved in flagellar structure and function, such as FlaA, FlgP, and Cjj0996 (Figure 6A), all of which are needed for cell invasion [27,28,43] but are increased to a more modest extent here. Other factors are likely to play a role, and we note that the phosphate transporter PstS is also enhanced over three-fold in the *rsmA* mutant, and this protein has previously been linked to *C. jejuni* infection of humans [44]. Similarly, the PstS protein of enterohemorrhagic *E. coli* (EHEC) and enteropathogenic *E. coli* (EPEC) supports adherence and colonization of intestinal epithelial cells [45]. Furthermore, this protein is also evident in Gram-positive bacteria, where the PstS homolog of *Mycobacterium tuberculosis* affords protection from the antimicrobial stress response in macrophages [46,47].

After the complementation of the *rsmA* mutant, the increased levels of most of the motility-related proteins fell to wild-type levels. However, the abundances of FlgE, FlgG, and FlaB fell below the original wild-type levels (Figure 6A) and were thus inversely proportional to the level of RsmA, which was over-expressed in the complemented strain. We note that the expression of *flgE*, *flgG,* and *flaB* is regulated by σ^54^ dependent promoters (Figure 6B).

As touched upon above, the 147 significant changes in protein levels detected in the comparison of the *rsmA* mutant and the wild-type strain could be substantially reduced by comparing the *rsmA* strain to its complementary derivative (Appendix A and Figure 6A). While this emphasizes the need for complementation to rule out as many effects as possible that are unrelated to the methyltransferase under investigation, this still leaves over a dozen changes in protein content that might not be a direct effect of RsmA activity. For instance, expression of the Cjj0003 gene downstream is severely downregulated, presumably as a polar effect of *rsmA* inactivation immediately upstream, and thus might not be expected to be rescued by reinstating *rsmA* at another site in the chromosome (Figure 1A). Cjj0003 encodes the 23S rRNA pseudouridine synthase RluB [48], and loss of this modification could also affect protein synthesis on the ribosome, which is the premise here for investigating the loss of RsmA modification.

RsmA is important for the maturation of the small ribosomal subunit [11], and its methylations, at a location close to the anti-Shine Dalgarno mRNA binding sequence at the 3′-end of 16S rRNA, undoubtedly play a role in translational initiation, as does the interaction of kasugamycin [49] at the same location [50]. However, from the initiation sites of the mRNAs most affected here (Appendix A), it is not immediately obvious what structural features might be responsible for altering their expression. Some changes in the proteome could result from collateral effects (such with Cjj0003) or as secondary effects from a regulator upstream in a biosynthetic pathway, such as FlhA (Figure 6B). The FlhA mRNA does, in fact, have an irregular Shine–Dalgarno structure (Appendix A).

Considering the numerous unintentional and unexpected changes that occur when manipulating gene expression in pathogens, an important question that must be addressed is whether any of the resultant strains becomes a more effective pathogen. Needless to say, the goal of the present study is to define the determinants involved in virulence and certainly not engage in gain-of-function to produce a more dangerous *C. jejuni* strain. Reassuringly, we can conclude that despite the enhanced motility, improved attachment, and invasive properties of the *rsmA* strain, it would make a very poor pathogen. Intuitively, the loss of a phylogenetically highly conserved gene such as *rsmA* must have negative physiological consequences. This is evident in other species, for instance, with growth defects at lower than optimal temperatures in an *E. coli rsmA* strain [11], and here also for the *C. jejuni rsmA* strain in a medium that is anything less than nutrient-rich (Figure 2D,E). In the wild, optimal temperature and nutrient conditions for growth, in the absence of competing strains, are never experienced, and the *rsmA* strain would be rapidly out-competed by other bacteria present, as demonstrated in competition growth experiments here (Figure 2F).

In conclusion, we have engineered a *C. jejuni* strain lacking RsmA activity and linked the resultant phenotypical defects with changes in the expression of specific proteins. Complementing this strain with an active copy of *rsmA* rescues most of these defects and, in addition, inadvertently overexpresses RsmA to cause other physiological and morphological changes. This study validates the premise that making subtle adjustments to the bacterial ribosome (in this case, interfering with one set of rRNA methylations) causes changes in the way mRNAs are translated, which in turn can have profound effects on bacterial virulence. Combining this with high-resolution proteomic analyses using mass spectrometry facilitates the correlation of phenotypical changes with the defective expression of specific proteins. Each of the numerous rRNA modifications, particularly the highly conserved ones, plays a role in the fine-tuning of the translation process, and (as in the case of RsmA and TlyA) these roles will be different. With over thirty such rRNA modifications in *E. coli* [6] and almost as many in *C. jejuni,* there are ample prospects for perturbing the infective cycles of bacterial pathogens to identify factors essential for virulence.

## 4. Materials and Methods

### 4.1. Bacterial Strains and Growth Conditions

*C. jejuni* strains used in this study (Table 2) were grown under microaerobic conditions at 37 °C on brain–heart infusion (BHI) agar containing 5% (*v*/*v*) sheep blood or BHI agar containing 5% (*v*/*v*) sheep blood supplemented with chloramphenicol at 20 µg mL^−1^ and/or kanamycin at 30 µg mL^−1^.

For growth assays in minimal medium, we used MCLMAN limited medium (Medium with Cysteine, Leucine, Methionine, Aspartate, and Niacinamide) [51] and the same medium without methionine (MCLAN). Growth assays were initiated from an OD_600_ of 0.02 after resuspended cells in the corresponding medium. Each strain was assayed three times in biological triplicate, with plates incubated under microaerophilic conditions at 37 °C for 72 h.

### 4.2. Construction of rsmA Knockout Mutant

The null mutant of the *C. jejuni* 81176 was created by inserting a chloramphenicol *cat* resistance cassette (Cm^r^) into *rsmA* gene. Briefly, the *rsmA* gene was amplified by PCR using the rsmAF: ATGGTTAAAGCAAAAAAACAATACGGAC and rsmAR: TTATTTATCTCTTTGTTTTCGTCCATATTTATC primers and cloned into the pJET1.2 plasmid (Thermo Fisher Scientific, Waltham, MA, USA), prior to insertion of the *cat* cassette [52] using the primers rsmA_mutR: AAAAGAAATGGCTGAAAAATTCTGCGC and rsmA_mutF: CATTACAATAAGCCCTAGACAATTTTTATC to disrupt the coding sequence replacing 30 bp of its sequence. The plasmid containing the engineered *rsmA* gene was used for electro-transformation of *C. jejuni* 81-176 cells. The plasmid lacks a *C. jejuni* replicon; therefore, retention of Cm^r^ required allelic exchange with the homologous chromosomal *rsmA* gene.

The resultant *C. jejuni* 81-176 Δ*rsmA* null mutant was complemented by inserting a wild-type copy of *rsmA* under control of its own promoter (with a total size of 2501 nucleotides) into the 121-bp intergenic region between *cjj0680* and *cjj0681* forming the *C. jejuni* strain 81-176 Δ*rsmA*::*rsmA* (Table 2; Figure 1A). All strains were verified by PCR and sequencing and frozen in aliquots immediately after construction to avoid any serial passaging before their use in the growth and virulence tests.

**Table 2 ijms-25-09797-t002:** Bacterial strains used in this study.

Strains	Relevant Characteristics	Source/Reference
*C. jejuni* 81-176	Wild-type	[53]
*C. jejuni* 81-176 Δ*rsmA*	Cm^r^, *rsmA* deletion mutant	This study
*C. jejuni* 81-176 Δ*rsmA*::*rsmA*	Cm^r^, Km^r^, *rsmA* deletion mutant complemented with *C. jejuni rsmA*	This study

### 4.3. Motility

The motility of *C. jejuni* cells was assessed by adding 3 µL of culture (OD_600_ = 0.5) onto BHI with 0.25% agar. Plates were left to dry and were incubated under microaerobic conditions for 48 h at 37 °C before measuring cell migration.

Direct real-time motility was observed and filmed using phase-contrast microscopy (Eclipse E400, Nikon, Tokyo, Japan). Bacteria were diluted to OD_600_ of 0.05–0.1 in BHI broth medium, 5 µL of culture were applied on a microscopy slide with coverslip, and motility was recorded immediately after preparation.

### 4.4. Virulence Phenotypes

Virulence characteristics of *C. jejuni* include its ability to form biofilms, its adhesion to and invasion of human epithelial cells, and its survival in macrophages as previously described [3]. Briefly, biofilm formation was measured spectroscopically after growing *C. jejuni* strains under microaerobic conditions in 24-well polystyrene plates. Biofilm formation was determined by crystal violet staining and measuring absorbance at 570 nm (A_570_) using a Bio-Mate spectrophotometer (Thermo Fisher Scientific, Waltham, MA, USA).

Invasion of *C. jejuni* was assayed using Caco-2 cells. Bacteria were added into the wells at an MOI of 100 bacteria to one epithelial cell. Infected monolayers were incubated for 2 h for invasion to occur. The monolayers were then washed three times with sterile phosphate-buffered saline (PBS) and incubated for 2 h in minimal essential medium containing 100 µg gentamicin mL^−1^ to kill extracellular bacteria. After this period, the monolayers were washed as described above and lysed with 0.1% Triton X-100 in PBS for 15 min at room temperature. Following serial dilution in PBS, invaded bacteria were enumerated by colony counting on BHI agar cultured under microaerophilic conditions.

For the survival assays, RAW 264.7 macrophages were incubated with *C. jejuni* at an approximate MOI of 100. Infected macrophage monolayers were incubated for 2 h, then medium containing 100 μg/mL of gentamicin was added to kill extracellular bacteria. The invasion period was monitored for 3, 6, 12, 24, and 48 h post-infection. Following each time point, the macrophages were lysed 0.1% Triton X-100 in PBS for 15 min at room temperature, and the live bacteria released were evaluated by plating serial dilutions on MH agar plates.

Statistical analysis was performed using IBM SPSS Statistics for Windows version 28 (IBM Corp., Armonk, NY, USA) software. The statistical significance of differences between tested groups was measured using the Mann–Whitney test.

### 4.5. MIC Determination

Overnight cultures of the *C. jejuni* strains were diluted to 0.5 McFarland standard and plated onto BHI agar containing sheep blood with kasugamycin concentrations increasing in two-fold steps. The MIC values are the lowest concentration at which no growth was observed after incubation under microaerobic conditions for 48 h at 37 °C.

### 4.6. Primer Extension

Total RNAs were extracted from *C. jejuni* strains by phenol extraction [54]. 5′-^32^P-end-labeled deoxynucleotide primers were hybridized to complementary regions of rRNA (16S nucleotides 1523-1540, *E. coli* numbering) and extended with AMV reverse transcriptase (Roche, Indianapolis, IN, USA). Extension products were run on denaturing polyacrylamide/urea gel and visualized by autoradiography using a Typhoon PhosphorImager (Amersham Bioscience, Amersham, UK) and identified by reference to dideoxynucleotide sequencing reactions run in parallel.

### 4.7. Competition Growth Assay and Florescence PCR

Approximate equal numbers of cells from two *C. jejuni* strains were mixed in either MH, MCLAN, or MCLMAN media with and without kasugamycin at 400 μg mL^−1^. This concentration of kasugamycin is defined here as a ‘sub-inhibitory’ level of the drug, which is less than that required to completely arrest growth (although growth will be slowed to different extents for the different strains). After a 24 h incubation cycle, cells were diluted 1000-fold, and the cycle was repeated. Aliquots of cells were harvested by centrifugation at the end of each cycle, and the cell pellets were resuspended in deionized water. The relative amounts of wild-type and mutated *rsmA* gene copies in the strain mixtures were assessed by Taq DNA Polymerase (Thermo Fisher Scientific, Waltham, MA, USA) PCR using the unlabeled forward primer (rsmAF) and the reverse primer (rsmAR) that was 5′-labeled with cyanine 3 (Cy3), both of which are complementary to regions of *rsmA*. The PCR products of 801 (for wild type) and 1184 base pairs (for the mutant with the *cat* insertion) were separated on agarose gels and visualized using the ChemiDoc Imaging System (Bio-Rad, Hercules, CA, USA). This approach has previously been used to follow competition between strains of *Mycobacterium smegmatis* in media with capreomycin [55].

### 4.8. Transmission Electron Microscopy (TEM)

The morphology of *C. jejuni* cells was evaluated by transmission electron microscope JEM-1220 (JEOL, Tokyo, Japan) after spreading droplets of bacterial culture onto TEM grids (Formvar on 3 mm 200 Mesh Cu Grids, Agar Scientific, Stansted, UK). Operating voltage was set to 80 keV.

### 4.9. Scanning Electron Microscopy (SEM)

Bacterial biofilms were grown on glass coverslips [2], and their morphology was examined using secondary electron (SE) and backscattered electron (BSE) images. SE images were generated using a Thermo Fisher Quattro S Environmental Electron Microscope (ESEM) at an accelerating voltage of 10 or 20 kV with working distance of approx. 10 mm; non-sputtered samples were investigated in low vacuum mode with an H_2_O chamber pressure of 50 Pa. Samples were either carbon-coated with Safematic CCU-010 or with a thin Pt layer with Bal-Tec SCD005 Sputter Coater.

### 4.10. Protein Extraction and Peptide Labeling

Overnight cultures of *C. jejuni* strains were harvested by centrifugation [2] and resuspended in lysis buffer (7 M urea, 2 M thiourea, 1% DTT, and 0.5% n-octyl-β-D-glucopyranoside) supplemented with protease inhibitors (Roche), and ultrasonicated for protein extraction. After measuring the protein concentrations (Bradford reagent, Merck, Rahway, NJ, USA), 200 µg samples were alkylated with 45 mM iodoacetamide (IAA), incubating for 30 min in the dark before adding 5 mM DTT and adjusting to pH 8.5 with 30 mM triethylammonium bicarbonate (TEAB) buffer. Proteins were predigested for 3 h at 37 °C with 0.005 AU lysyl endopeptidase (Lys-C, FUJIFILM Wako Pure Chemical Corporation, Hong Kong, China). Samples were then digested further at 37 °C with methylated trypsin at an enzyme/protein ratio of 1:50 [56] prior to quenching with 10% trifluoroacetic acid (TFA) and centrifuging at 20,000× *g* for 15 min. Peptides were desalted on R2C8 and R3C18 resin reversed-phase columns (Thermo Fisher Scientific), washed with 0.1% TFA, and eluted with 60% acetonitrile/0.1% TFA before drying under vacuum centrifugation and resuspending in 50 mM TEAB buffer.

The peptides were labeled with isobaric tags (TMTpro, Thermo Fisher Scientific, Waltham, MA, USA) according to the supplier’s recommendations. Based on total peptide intensity in the different samples, estimated in test runs, the samples were mixed in 1:1 (*w*/*w*) proportions and desalted on reverse-phase columns.

### 4.11. LC-MS3 Tandem Mass Spectrometry

The TMTpro-labeled peptides were fractionated into 12 concatenated fractions by reversed-phase chromatography in high pH buffers on a Dionex UltiMate 3000 (Thermo Fisher Scientific) equipped with a nanoEasy M/Z Peptide, CSH C18 130 Å, 1.7 µm, 300 µm × 100 mm analytical column (Waters, Milford, CT, USA). Peptides in the mobile phase of 20 mM ammonium formate in water, pH 9.3, were eluted at a flow rate of 5 µL/min over 71 min in a gradient of solvent B (80% acetonitrile and 20% solvent A) rising from 2% to 40%, followed by 50% solvent B to 122 min, before finally washing with 95% solvent B for 10 min. and then drying.

Fractions were dissolved in 5 μL solvent A (0.1% formic acid in water) and separated on an in-house prepared 20 cm analytical columns (100 μm ID, pulled tip fused silica capillary, containing 1.9 μm Reprosil-Pur 120 C18-AQ, Dr. Maisch GmbH, Ammerbuch, Germany). Peptides were separated at a flow rate of 300 μL/min with increasing concentrations of Solvent B (90% acetonitrile, 0.1% formic acid) beginning at 2% and raising to 8% over 2 min, then to 22% over the next 60 min, to 40% in 10 min, to 95% in 1 min and holding at 95% for 5 min, and finally to 100% over 1 min and holding at 100% for 3 min.

Mass spectrometry analyses were performed using an SPS-MS3 approach on an Orbitrap Eclipse Tribrid mass spectrometer (Thermo Fisher Scientific). Full scans over a mass range of 350–1600 *m*/*z* were performed within 3 s cycles at a resolution of 120 k in positive ion mode. Subsequently, precursor ions at 400–1600 *m*/*z* were selected for CID fragmentation and detection in Ion Trap (NCE 35, CID activation time 10 sec, Ion Trap Scan Rate 10 ms). MS2 spectral data were screened against the *Campylobacter* protein database (fasta file from UniProt-the Universal Protein Resource for protein sequence) with a search time limit of 35 ms. Static modifications were carbamidomethyl (C) and TMTpro 16plex (N-terminus, K), with a single missed cleavage being allowed. SPS-MS3 fragmentation was triggered after positive identification of precursor ions: 10 SPS precursor ions were accumulated and further fragmented with HCD (NCE 55, resolution 30k, scan range 100–500 *m*/*z*, maximum injection time “Auto”) and detected by Orbitrap for sample quantification.

### 4.12. Protein Identification and Quantification

Raw data files were processed by Thermo Proteome Discoverer software v 2.5 (Thermo Fisher Scientific). Protein identification was made from the *C. jejuni* 81-176 genome sequence (UniProt databases) using the Sequest search engine. For the search parameters, precursor mass tolerance was set to 10 ppm, with a fragment mass tolerance of 0.6 Da; carbamidomethylation of cysteine and TMTpro modification of N-termini and lysine residues were added to dynamic modifications, while acetylation was set to variable modifications; and two missed cleavage sites per protein were allowed. For reporter ion quantification, the co-isolation threshold was set to 50, and average reporter ion S/N threshold to 10. Data from Proteome Discoverer were processed using Excel 365 (Microsoft version 2408).

Quantification was based on two biological replicates of each strain and taking into consideration only the proteins that were identified from at least two unique peptides and with a false discovery rate of less than 1%. The abundances of individual proteins that differed with *p*-values below 0.05 in comparisons of the wild-type strain with the Δ*rsmA* strain and Δ*rsmA* with the complemented Δ*rsmA*::*rsmA* strain were considered significant.

## Figures and Tables

**Figure 1 ijms-25-09797-f001:**
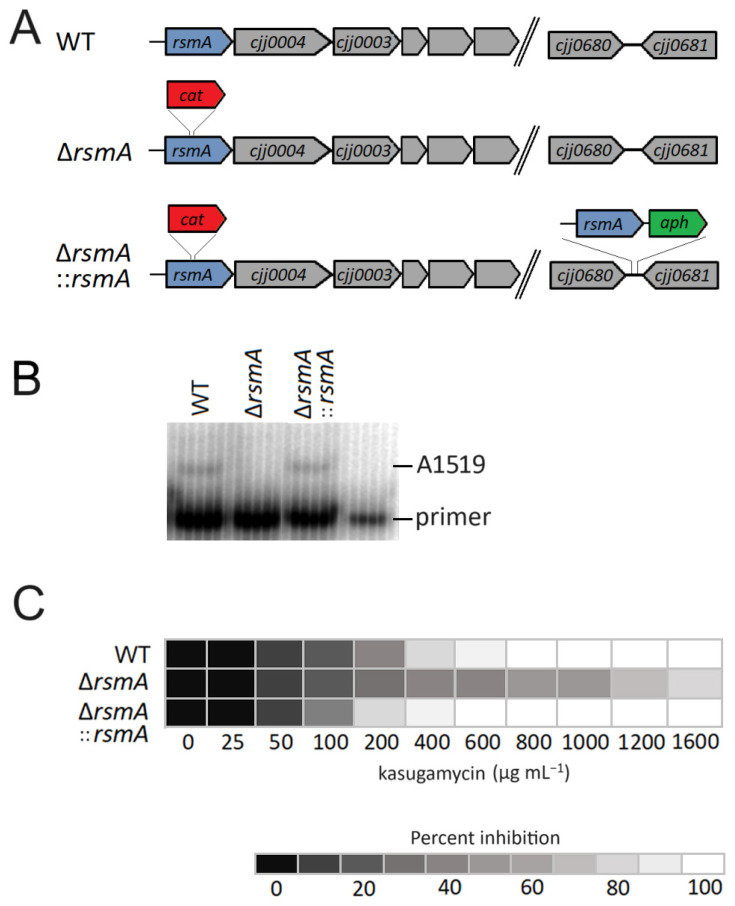
Inactivation of *rsmA* in the *C. jejuni* chromosome. (**A**) *C. jejuni* 81-176 *rsmA* gene locus. The Δ*rsmA* mutant was constructed from the wild-type (WT) strain by deleting 10 bp of *rsmA* and inserting the *cat* (Cm^R^) cassette at the same site. In the complemented strain, Δ*rsmA::rsmA*, an active copy of the *rsmA* gene under its own promoter, was inserted together with the *aphA* (Km^R^) cassette at the intergenic region between *cjj0680* and *cjj0681* (lower right)*.* (**B**) Gel autoradiograms of primer extensions on 16S rRNA from WT (wild-type) strain, the Δ*rsmA* mutant strain, and the Δ*rsmA*::*rsmA* strain complemented with an active copy of *rsmA.* (**C**) Kasugamycin MIC determination in liquid cultures for WT, the *rsmA* mutant, and the complemented strains.

**Figure 2 ijms-25-09797-f002:**
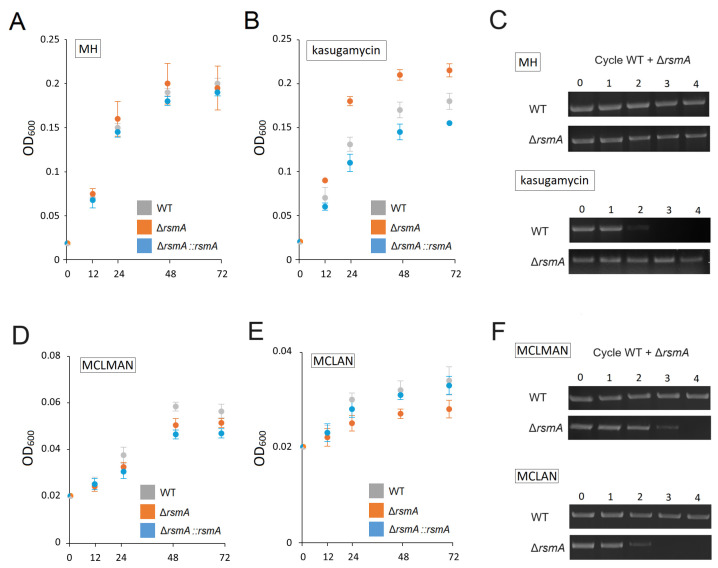
Growth of *C. jejuni* strains in liquid medium. (**A**) Strains grown in MH broth at 37 °C. All strains started from an initial optical density (OD) of 0.02 and were measured after 12, 24, 48, and 72 h. There was no significant difference in the growth of strains in MH broth. (**B**) Strains grown in MH broth with a sub-inhibitory concentration of kasugamycin (400 µg mL^−1^): *p* < 0.005 for WT versus Δ*rsmA* and *p* < 0.05 for Δ*rsmA* versus Δ*rsmA::rsmA*. (**C**) Growth competition assay. The WT and Δ*rsmA* strains were grown together through four 24 h cycles in MH broth with and without kasugamycin at 400 µg mL^−1^. The proportion of each strain was determined by PCR with one of the primers fluorescently labeled. Upper row, no drug; lower row, with kasugamycin. Each data set is representative of a minimum of three growth assays. (**D**) Strains grown in the minimal medium MCLMAN: after 24, 48, and 72 h, *p* < 0.001 for WT versus Δ*rsmA*; no significant difference was observed for Δ*rsmA* versus Δ*rsmA::rsmA*. (**E**) Strains grown in the MCLAN medium: after 24, 48, and 72 h, *p* < 0.001 for WT versus Δ*rsmA* and *p* < 0.05 for Δ*rsmA* versus Δ*rsmA::rsmA*. (**F**) Cell growth competition assay with WT and Δ*rsmA* strains grown together through four cycles in MCLMAN and MCLAN. Each data set is representative of a minimum of three growth assays.

**Figure 3 ijms-25-09797-f003:**
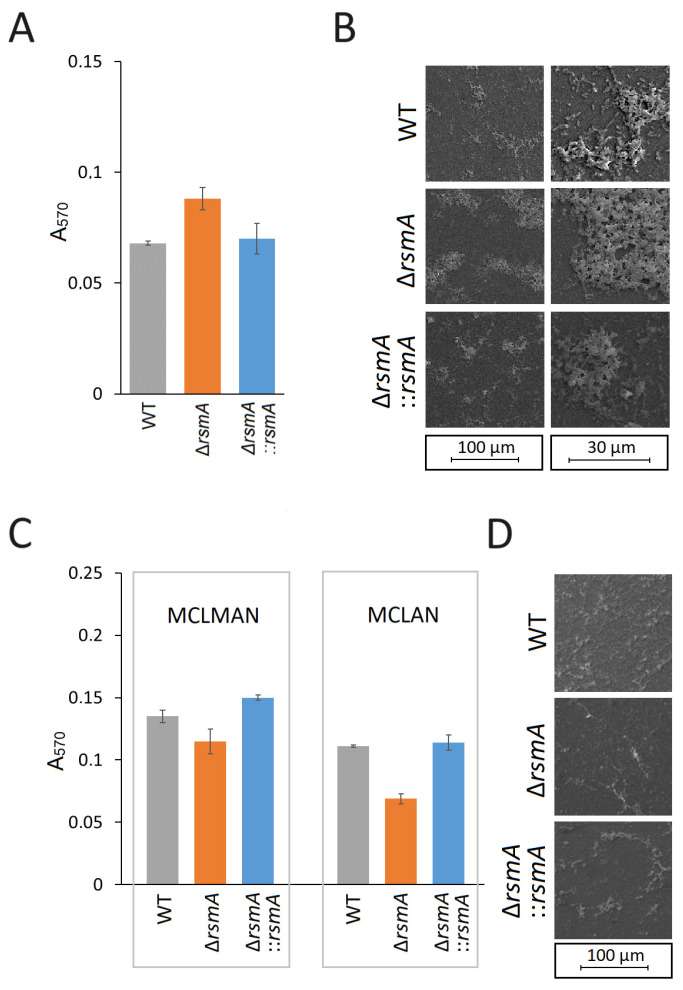
Effects of *rsmA* inactivation on *C. jejuni* biofilm formation at 37 °C. (**A**) Biofilm formed on polystyrene surfaces after 48 h, quantified by crystal violet staining (color-coded as in Figure 2); *p* < 0.05 for wild-type (WT) versus Δ*rsmA* and *p* < 0.05 for Δ*rsmA* versus Δ*rsmA*::*rsmA*. (**B**) Biofilm produced by *C. jejuni* on cover glass after 48 h under microaerobic conditions visualized by Field-Emission Scanning Electron Microscopy. (**C**) Biofilms of *C. jejuni* strains formed in the minimal media MCLMAN and MCLAN after 48 h. In MCLMAN, *p* < 0.01 for WT versus Δ*rsmA*; *p* < 0.001 for Δ*rsmA* versus Δ*rsmA*::*rsmA*. In MCLAN, *p* < 0.001 for WT versus Δ*rsmA* and for Δ*rsmA* versus Δ*rsmA*::*rsmA*. Values represent means ± S.E.M. of three independent experiments. (**D**) Biofilm produced by *C. jejuni* in MCLMAN on cover glass after 48 h under microaerobic conditions visualized by Field-Emission Scanning Electron Microscopy; biofilms formed under the same conditions in MCLAN remained scanter for all three strains.

**Figure 4 ijms-25-09797-f004:**
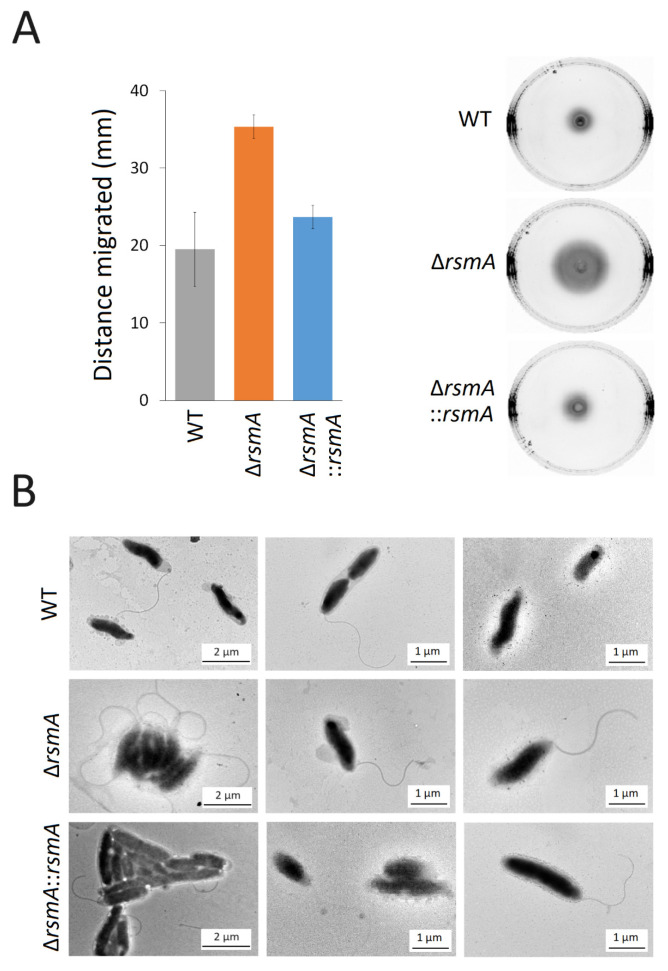
Motility of the *C. jejuni* strains. (**A**) Motility of the *C. jejuni* strains on agar plates after 48 h growth. Strain motility is summarized in the histogram; values represent the means ± SEM of three independent experiments. Significant differences were observed for migration of the wild-type (WT) strain compared to the Δ*rsmA* null strain (*p* < 0.005) and for the Δ*rsmA* versus the complemented Δ*rsmA*::*rsmA* strain (*p* < 0.005). There was no significant difference between the wild-type and the complemented strains. (**B**) Representative Transmission Electron Micrographs of *C. jejuni* strains showing cell morphology and flagella.

**Figure 5 ijms-25-09797-f005:**
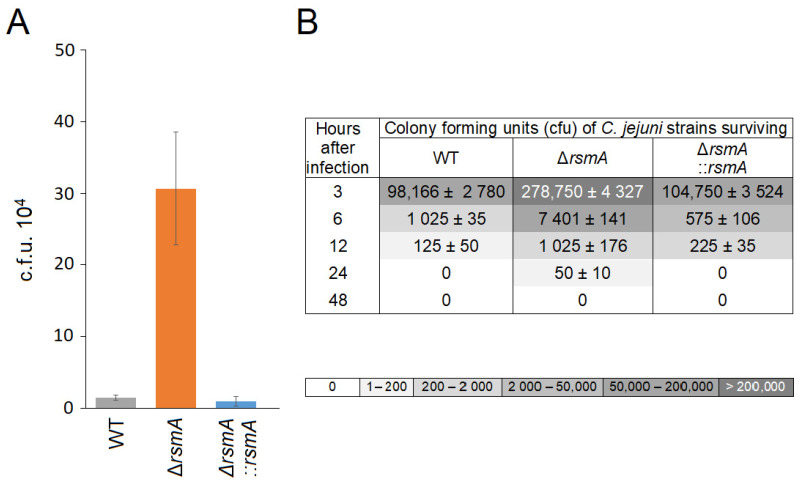
Invasive properties of the *C. jejuni rsmA* mutant. (**A**) Invasion of *C. jejuni* into Caco-2 epithelial cells was significantly increased by *rsmA* inactivation; *p* < 0.0001 for WT versus Δ*rsmA*, and also *p* < 0.0001 for Δ*rsmA* versus Δ*rsmA*::*rsmA.* (**B**) Survival of *C. jejuni* strains within macrophage RAW264.7 cells. The macrophages were infected with 10^7^ cfu of the *C. jejuni* strains (time zero). Viable intracellular *C. jejuni* cells are tabulated with shading to indicate >200,000, 50,000 to 200,000, 2000 to 50,000, 200 to 2000, and 20 to 200 surviving cells over 24 h. No viable *C. jejuni* cells were detected at 48 h. *p* < 0.001 for the wild-type (WT) strain versus the Δ*rsmA* strain; *p* < 0.01 for Δ*rsmA* versus Δ*rsmA*::*rsmA*. The cfu values are means ± SEM of three independent experiments.

**Figure 6 ijms-25-09797-f006:**
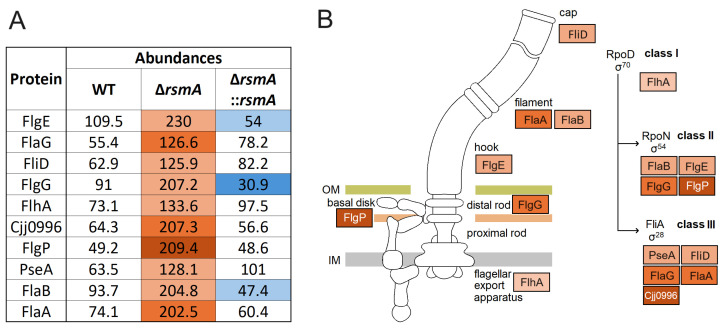
Structure of the *C. jejuni* flagella. (**A**) Significant changes in the expression of *C. jejuni* flagellar assembly components after *rsmA* deletion. The abundances of upregulated and downregulated proteins, compared with the wild-type strain, are marked in shades of red and blue, respectively. Further details can be found in Table 1 and Appendix A. (**B**) KEGG pathway for flagellar assembly in *C. jejuni*. Only proteins with significant changes (*p* < 0.05) are indicated. Schematic of the *C. jejuni* flagellum and its sequential assembly, adapted from the KEGG website (https://www.genome.jp/kegg/pathway.html (accessed on 25 November 2020)).

**Table 1 ijms-25-09797-t001:** Significant differences in the proteome of the *C. jejuni* wild-type strain (WT) caused by inactivation of *rsmA* and those that are rescued by complementation with an active copy of *rsmA*.

Description	Protein	Gene ID	Abundance Ratio	
			Δ*rsmA*/ WT	Δ*rsmA*/ Δ*rsmA*::*rsmA*
Flagellar hook protein	FlgE	*CJJ81176_0025*	2.1	4.2
Flagellar protein FlaG	FlaG	*CJJ81176_0572*	2.2	1.6
Flagellar hook-associated protein	FliD	*CJJ81176_0573*	2.0	1.5
Flagellar basal-body rod protein	FlgG	*CJJ81176_0721*	2.2	6.6
Flagellar biosynthesis protein	FlhA	*CJJ81176_0890*	1.8	1.3
PaaI family thioesterase	Cjj0996	*CJJ81176_0996*	3.2	3.6
Putative lipoprotein.	FlgP	*CJJ81176_1045*	4.2	4.3
Flagellin modification protein	PseA	*CJJ81176_1333*	2.0	1.2
Flagellin	FlaB	*CJJ81176_1338*	2.1	4.3
Flagellin	FlaA	*CJJ81176_1339*	2.7	3.3
30S ribosomal protein S12	RpsL	*CJJ81176_0511*	2.2	1.5
Putative imidazole glycerol phosphate synthase subunit	HisF2	*CJJ81176_1331*	2.1	1.4
Imidazole glycerol phosphate synthase subunit	HisH1	*CJJ81176_1332*	2.9	1.4
Amino acid-binding protein	GlnH	*CJJ81176_0836*	0.5	0.4
Phosphate ABC transporter. periplasmic phosphate-binding protein	PstS	*CJJ81176_0642*	3.0	1.3
Thioredoxin family protein	Cjj1656	*CJJ81176_1656*	1.9	1.4
Peptidase. M24 family	PepP	*CJJ81176_0681*	1.9	1.9
Peptidase. M23/M37 family	Cjj1105	*CJJ81176_1105*	1.8	1.3
Pseudouridine synthase	RluB	*CJJ81176_0003*	0.4	0.7
pTet and pVir proteins				
Uncharacterized protein	Cpp33	*CJJ81176_pTet0030*	2.6	1.4
Single-stranded DNA-binding protein	Ssb	*CJJ81176_pTet0031*	2.1	1.3
Uncharacterized protein	Cpp35	*CJJ81176_pTet0032*	2.1	1.3
VirB8	VirB8	*CJJ81176_pVir0001*	0.5	0.6
VirB9	VirB9	*CJJ81176_pVir0002*	0.5	0.6
Uncharacterized protein	Cjp07	*CJJ81176_pVir0008*	0.5	0.5
Uncharacterized protein	Cjp08	*CJJ81176_pVir0009*	0.4	0.6
VirB4	VirB4	*CJJ81176_pVir0053*	0.5	0.6
Uncharacterized protein	Cpp20	*CJJ81176_pTet0015*	0.4	0.6
Uncharacterized protein	Cpp47	*CJJ81176_pTet0044*	0.4	0.5
Ribosomal RNA small subunit methyltransferase A	RsmA	*CJJ81176_0005*	0.2	0.08

Upregulated and downregulated proteins with significant changes of *p* > 0.05 are marked in red and blue, respectively.

## Data Availability

The mass spectrometry proteomics data have been deposited to the ProteomeXchange Consortium via the PRIDE [57] partner repository with the dataset identifier PXD052567.

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
