# Peer review of "Increased Motility in Campylobacter jejuni and Changes in Its Virulence, Fitness, and Morphology Following Protein Expression on Ribosomes with Altered RsmA Methylation"

_ijms, 2024, doi:10.3390/ijms25189797_

Round 1

Reviewer 1 Report

Comments and Suggestions for Authors

Minor comments:

Maintain the same font style and size throughout the manuscript.

Line 93: ‘Consistent with this, growth of the null-strain in medium with sub-inhibitory concentration  (400 μg ml-1) of kasugamycin was faster than the wild-type and complemented strains’. To support this, authors should add the respective growth curves in place of figure 2a and also 2C.

Line 150: Authors should add the Field Emission Scanning Electron Micrograph for the Biofilm produced in MCLAN growth medium for comparison.

Line 165 and Figure 5A: Authors should clearly mention the time point of measurement. The authors should add a similar table as 5B.

The authors have not mentioned which statistical tests and software were used.

Supplementary TS1 is missing.

Author Response

Comments 1: Maintain the same font style and size throughout the manuscript.

Response: Font size and type corrected throughout.

Comments 2: Line 93: ‘Consistent with this, growth of the null-strain in medium with sub-inhibitory concentration  (400 μg ml-1) of kasugamycin was faster than the wild-type and complemented strains’. To support this, authors should add the respective growth curves in place of figure 2a and also 2C.

Response: Growth curves are now shown in Fig. 2A, 2B, 2D and 2E. A clearer definition of ‘sub-inhibitory concentration’ has been added to the M&M section (ll. 468-471).

Comments 3: Line 150: Authors should add the Field Emission Scanning Electron Micrograph for the Biofilm produced in MCLAN growth medium for comparison.

Response: The strains formed very scant biofilms on the MCLAN medium, as clearly shown with crystal violet staining (Fig. 3C). Showing these micrographs would not add anything of value to the manuscript (a note to this effect has been added to the Fig. 3 legend, l. 152). The A570 values on the histograms have been corrected.

Comments 4: Line 165 and Figure 5A: Authors should clearly mention the time point of measurement. The authors should add a similar table as 5B.

Response: Fig. 5A shows an invasion test of Caco-2 cells, and the number of invaded C. jejuni cells were measured after 4 h total from infection. A clearer account of this assay is now in M&M (ll. 436-443).

Figure 5B shows survival in macrophages, the number of C. jejuni cells were measured at intervals, as shown in the table in the figure. We have revised the description of the assay in M&M (ll. 444-450) to fill in any missing details.

Comments 5: The authors have not mentioned which statistical tests and software were used.

Response: The statistical tests and software that were used have been added to M&M section 4.4, ll. 451-3.

Comments 6: Supplementary TS1 is missing.

Response: Table S1 was submitted as a separated Excel file.

Also, italics removed where necessary.

Final note: We are grateful to the two reviewers for giving their comments and insight into our study. We believe that taking these into account has significantly improved the manuscript.

Reviewer 2 Report

Comments and Suggestions for Authors

Manuscript by Agnieszka Sałamaszyńska-Guz et al., titled "Increased motility in Campylobacter jejuni and changes in its virulence, fitness, and morphology following protein expression on ribosomes with altered RsmA methylation". This manuscript focuses on the role of RsmA (rRNA methyltransferase) in the virulence of C. jejuni. However, it needs to clarify the following concerns:

Major concerns

-The cell growth of C. jejuni strains seems to be slower than expected. According to the methods and materials section (M&M), the initial inoculum is 0.05 OD600. Figure 2A shows that the OD after 48 hours is only about 0.18. This growth rate is much lower than the expected range of 0.5 to 0.8 for C. jejuni 81-176 in MH media (refer to this article: doi: 10.1128/JB.01222-09). Additionally, Figure 2C clearly indicates that there is no growth at all in the minimal media with an initial inoculum of 0.05 (as described in M&M).

-Cell growth competition assay; Figures B & D, authors said “ in direct competition assays with this amount of drug, the wild-type cells were no longer evident after one growth cycle (Fig 2B) “. I do not believe this to be the case after one growth cycle, and most likely the natural transformation between two strains occurs where the mutated gene transformed to WT under the stress of kasugamycin. The authors should test both strains with these specific primers to verify this. Also, I could not find the result for the complemented strain.

-Motility assay could give misleading results due to phase variation after transformation. The hyper-motile phenotype usually appears after transformation. Could you confirm that steps after the transformation, more details, in section 4.2

Minor

- Clarify the size of each PCR fragment in Figure 1B.

- Move Figure 6 from the discussion to the results.

- Remove the italics from "infection" on line 16 and remove the word "now" from line 34.

- Verify the font size and type throughout the entire manuscript (e.g., Line 39-40, 64-66, etc.).

- Move Table 2 to supplementary data.

- Add more details to the Materials and Methods section, including the size of the rsmA gene amplified by PCR and a brief description of all protocols used in section 4.4.

- Ensure that images in Figure 4A are of high resolution.

- Review all references for inconsistent capitalization.

Author Response

Manuscript by Agnieszka Sałamaszyńska-Guz et al., titled "Increased motility in Campylobacter jejuni and changes in its virulence, fitness, and morphology following protein expression on ribosomes with altered RsmA methylation". This manuscript focuses on the role of RsmA (rRNA methyltransferase) in the virulence of C. jejuni. However, it needs to clarify the following concerns:

Major concerns

Comments 1: The cell growth of C. jejuni strains seems to be slower than expected. According to the methods and materials section (M&M), the initial inoculum is 0.05 OD600. Figure 2A shows that the OD after 48 hours is only about 0.18. This growth rate is much lower than the expected range of 0.5 to 0.8 for C. jejuni 81-176 in MH media (refer to this article: doi: 10.1128/JB.01222-09). Additionally, Figure 2C clearly indicates that there is no growth at all in the minimal media with an initial inoculum of 0.05 (as described in M&M).

Response: Thank you for drawing our attention to this point. The value in M&M was a typo, and is now corrected (I. 399). The initial inoculum was 0.02, as also shown in the new version of Fig. 2. We agree that the growth rate of our 81-176 strain is nevertheless slightly slower than that observed by Naito et al. (2010). However, all our derivative strains were constructed from our 81-176 strain, and therefore all the data presented here are mutually consistent.

Comments 2: Cell growth competition assay; Figures B & D, authors said “in direct competition assays with this amount of drug, the wild-type cells were no longer evident after one growth cycle (Fig 2B)“. I do not believe this to be the case after one growth cycle, and most likely the natural transformation between two strains occurs where the mutated gene transformed to WT under the stress of kasugamycin. The authors should test both strains with these specific primers to verify this.

Response: This part of the text did indeed require a more precise analysis and description. Zooming in on Fig. 2C, a faint band can still be discerned for the WT strain in kasugamycin after the second cycle. The text has been revised (l. 97) to reflect this. Inoculated as pure cultures in kasugamycin (new Fig. 2B), the WT strain (doubling time approx. 7½ h) was initially growing about 50% slower than the rsmA strain (doubling time approx. 5h). The number of rsmA cells would be appreciably greater than that of the WT strain after two growth periods of 24 h (we estimated there would be around 15-times more rsmA cells, and this difference would presumably be greater when grown in direct competition in the same medium, with competition for available nutrients). Our present interpretation is consistent with the growth data for the strains in isolation and in competition, and we believe there is no need to invoke additional explanations such as genetic exchange during the course of the assay.

Comments: Also, I could not find the result for the complemented strain.

Response: The complemented strain was inhibited to an even greater extent by the presence of kasugamycin (this is clearly shown in Fig. 1C and now in the new version of Fig. 2B).

Comments 3: Motility assay could give misleading results due to phase variation after transformation. The hyper-motile phenotype usually appears after transformation. Could you confirm that steps after the transformation, more details, in section 4.2

Response: True - if phase variation had occurred it could indeed lead to misinterpretation of the data – but we ensured that it did not occur. Motility and other phenotypic traits were assayed in all cases without any serial passaging of the strains after their construction (a note to this effect added in Section 4.2, ll. 417-419). Although we don’t offer this as a conclusive rebuttal, phase variation might be expected first after a few in vitro passages of the strain. A more conclusive argument against the occurrence of phase variation is the lack of any change in the amounts of proteins, such as MotA, FlgR, FlgS (Supplementary Table S1), that are associated with motility and subject to phase variation. Importantly, motility returned to wild-type levels after complementation – and would not have been rescued if it were due to phase variation.

Minor

Comments 4: Clarify the size of each PCR fragment in Figure 1B.

Response: This is a primer extension assay with reverse transcriptase. When A1519 is dimethylated, extension of the primer is arrested after three nucleotides.

Comments 5: Move Figure 6 from the discussion to the results.

Response: Done as requested

Comments 6: Remove the italics from "infection" on line 16 and remove the word "now" from line 34.

Response: Done.

Comments 7: Verify the font size and type throughout the entire manuscript (e.g., Line 39-40, 64-66, etc.).

Response: Done

Comments 8: Move Table 2 to supplementary data.

Response: A comprehensive list of the proteomics data is presently in supplementary information section (Table S1). For the reader’s easy of following the relevant changes in protein abundances, we prefer to keep Table 2 in the Results section.

Comments 9: Add more details to the Materials and Methods section, including the size of the rsmA gene amplified by PCR and a brief description of all protocols used in section 4.4.

Response: The rsmA gene size added in the complemented strain is now given on l. 415. PCR fragment sizes in the assay for survival in the competition assays are given on ll. 477-8. A fuller account of the methods has been added to section 4.4. (ll. 430-453) as requested.

Comments 10: Ensure that images in Figure 4A are of high resolution.                

Response: High resolution images are provided.

Comments 11: Review all references for inconsistent capitalization.

Response: References are now good.

Also, italics removed where necessary.

Final note: We are grateful to the two reviewers for giving their comments and insight into our study. We believe that taking these into account has significantly improved the manuscript.

Round 2

Reviewer 2 Report

Comments and Suggestions for Authors

I am dissatisfied with the author's response to my comments. I strongly recommend repeating the growth assays. It is clear to me that the OD values are too low and inconsistent with the biofilm experiments. For instance, the OD for the biofilm assays after 48 hours in MCLMAN and MCLAN is above 0.1, despite no growth being visible in Figures 2D and E (~0.06 at 72 hours). The authors must provide the number of cells (CFU/mL) and the optical density (OD) for both the growth MH and growth competition experiments and include the data in supplementary materials. Also, the author mentioned, “Motility and other phenotypic traits were assayed in all cases without any serial passaging of the strains after their construction. (also, Line 413-415)” However, phase variation can occur randomly following transformation, especially in Campylobacter. One fundamental step in avoiding such effects after producing mutants and complemented mutants is to correct the background for those strains via natural transformation using WT before analyzing the phenotypes. Therefore, I do not recommend this work be published in its current form.

Author Response

Comments: I am dissatisfied with the author's response to my comments. I strongly recommend repeating the growth assays. It is clear to me that the OD values are too low and inconsistent with the biofilm experiments. For instance, the OD for the biofilm assays after 48 hours in MCLMAN and MCLAN is above 0.1, despite no growth being visible in Figures 2D and E (~0.06 at 72 hours).

Response: There has been a series of misunderstanding here, and also in the following comments from the reviewer. We will attempt to resolve them below.

Figure 2 records the growth of planktonic cells in liquid culture. The optical densities were recorded at 600 nm, and determine the light scattering by cells in suspension in the medium, which in turn enables us to follow the planktonic growth of the cells.

On the other hand, Figure 3 measures cells that are no longer in suspension, but have formed immobile biofilms on a plastic surface. These measurements were taken after the cells had been stained with crystal violet, and were made at 570 nm (the standard wavelength for determining absorbance using this stain).

The two approaches and measurements are fundamentally different and cannot be compared as the reviewer suggests.

Comments: The authors must provide the number of cells (CFU/mL) and the optical density (OD) for both the growth MH and growth competition experiments and include the data in supplementary materials.

Response: The competition assay and the method of measurement we have employed are designed to follow the competitive growth of two strains in the same medium. The assays begin at time zero with equal numbers of cells from each strain, and then determines how this relative proportion of 1:1 changes over several growth cycles. The relevant point here is the changing proportion of the cell strains, which indicates their relative fitness under the culture conditions employed. The series of experiments presented in the manuscript clearly does precisely this. This is an established technique and we have now added a reference to a previous application of the method (ll. 475-7). Measurements of the number of cells (CFU/mL) and the optical density (OD) are irrelevant for this assay, but have been made where necessary in other contexts in the manuscript (e.g. Fig. 2A, 2B and 5B).

Comments: Also, the author mentioned, “Motility and other phenotypic traits were assayed in all cases without any serial passaging of the strains after their construction. (also, Line 413-415)” However, phase variation can occur randomly following transformation, especially in Campylobacter. One fundamental step in avoiding such effects after producing mutants and complemented mutants is to correct the background for those strains via natural transformation using WT before analyzing the phenotypes.

Response: This comment would have been more helpful if the reviewer had documented with a reference their assertion of a connection between motility and phase variation that occurs ‘especially in Campylobacter’.  Whether or not such documentation exists (we could not find any), the fact remains that we have controlled for any such eventualities by rescuing the motility phenotype by adding an active copy of the rsmA gene back to the null-mutant. This in itself shows that the motility change is directly connected with RsmA function.